# Municipal Ethnic Composition and Disparities in COVID-19 Infections in New Jersey: A Blinder–Oaxaca Decomposition Analysis

**DOI:** 10.3390/ijerph192113963

**Published:** 2022-10-27

**Authors:** Yuqi Wang, Laurent Reyes, Emily A. Greenfield, Sarah R. Allred

**Affiliations:** 1Department of Social Work, China Youth University of Political Studies, Beijing 100089, China; 2School of Social Welfare, University of California, Berkeley, CA 94720, USA; 3School of Social Work, Rutgers University, New Brunswick, NJ 08901, USA; 4Department of Psychology, Rutgers University, Camden, NJ 08102, USA

**Keywords:** COVID-19, racial/ethnic disparity, health disparity, infection rate, Latinx

## Abstract

COVID-19 has disproportionally impacted Latinx and Black communities in the US. Our study aimed to extend the understanding of ethnic disparities in COVID-19 case rates by using a unique dataset of municipal case rates across New Jersey (NJ) during the first 17 months of the pandemic. We examined the extent to which there were municipal-level ethnic disparities in COVID-19 infection rates during three distinct spikes in case rates over this period. Furthermore, we used the Blinder–Oaxaca decomposition analysis to identify municipal-level exposure and vulnerability factors that contributed to ethnic disparities and how the contributions of these factors changed across the three initial waves of infection. Two clear results emerged. First, in NJ, the COVID-19 infection risk disproportionally affected Latinx communities across all three waves during the first 17 months of the pandemic. Second, the exposure and vulnerability factors that most strongly contributed to higher rates of infection in Latinx and Black communities changed over time as the virus, alongside medical and societal responses to it, also changed. These findings suggest that understanding and addressing ethnicity-based COVID-19 disparities will require sustained attention to the systemic and structural factors that disproportionately place historically marginalized ethnic communities at greater risk of contracting COVID-19.

## 1. Introduction

The COVID-19 pandemic has increased attention to health inequity and structural racism worldwide. As the US Centers for Disease Control and Prevention (CDC) summarized, the pandemic “…has highlighted that health equity is still not a reality as COVID-19 has unequally affected many racial and ethnic minority groups, putting them more at risk of getting sick and dying from COVID-19” [1]. Structural racism is one of the most widespread forms of oppression today and a major driver of health disparities globally [1,2,3]. We define structural racism as the mechanisms by which social, political, and economic systems (i.e., housing, education, transportation, employment, and criminal justice) are structured to privilege and protect white supremacy through the ongoing oppression and persecution of Black and Indigenous people, as well as people from the Global South [4,5]. The negative health outcomes associated with racism have been well documented in the literature, showing health disparities across physical and mental health, biological measures, and the risk of viral infection [1,2,3,6,7,8].

Thus far, there are numerous studies showing increased rates of COVID-19 spread, hospitalizations, and mortality in the US among Black and Latinx populations compared to the white population. The CDC data demonstrate that COVID-19 cases are highest amongst Latinx people than any other racial group, and they are twice as likely to die from COVID-19 and three times more likely to be hospitalized than white non-Hispanic individuals [9]. In a systematic review of 54 peer-reviewed observational studies on racial/ethnic and socioeconomic disparities regarding COVID-19, Khanijahani and colleagues concluded that, despite inconsistencies across the measures and research designs, most studies demonstrated “that racial/ethnic minority populations in a region are at greater risk of COVID-19 infection, hospitalization, confirmed diagnosis, and mortality [10]”.

Our study aims to extend the empirical evidence on ethnic disparities in COVID-19 health outcomes in two ways. First, using a unique dataset with municipal case rates in New Jersey (NJ), we aimed to replicate the findings from prior research indicating an elevated risk of COVID-19 infection among communities with high proportions of residents identified as Latinx and/or Black. Second, guided by the World Health Organization’s (WHO) Health Inequality Casual Model [11] and using the Blinder–Oaxaca decomposition analysis [12,13], we explore the extent to which municipal-level vulnerability and exposure factors account for disparities by ethnic composition, focusing on the potential differences at different points during the first 17 months of the pandemic in NJ (during which the Alpha variant was predominant).

Throughout our study, we use the term “ethnic” in place of “racial” because we understand the term *race* as stemming from notions of inherent biological markers that separate individuals into groups based on skin color or physical features, which further reinforces racist ideologies [14]. Instead, the term *ethnicity* acknowledges differences and similarities in ancestry, history, language, customs, and beliefs that can be held within one individual and may differ between two people of the same skin color [14]. However, we maintain the use of the term racism, as a systemic, structural, and individual practice that assumes people’s worth and dignity by their proximity to whiteness.

## 2. Empirical Foundation

### 2.1. Ethnic Disparities in COVID-19 Infection Based on Geospatial Data

Most US epidemiological studies of COVID-19 based on geospatial data, whereby parcels of land are the focal unit of analysis, have used state- and county-level data, given that publicly available datasets typically report data at these levels. (For an overview of counties as a geographic unit in the U.S., see the National Association of Counties [NACO] [15].) Studies using state and county data have found that geographic locations with larger percentages of individuals from Black and Latinx communities have higher rates of COVID-19 infection and mortality [16,17,18,19]. For example, Millet and colleagues found that counties with a Black population greater than 13% accounted for over half of the COVID-19 cases and deaths nationwide during the very first part of the COVID-19 pandemic (up to 13 April 2020) [17]. From 677 counties with a >13% Black population, 656 (90%) reported a COVID-19 case [17] compared to 81% among counties with <13% of Black residents [16]. In addition, analyses of public health data in major U.S. metropolitan areas, such as New York City (NYC) and Chicago [16,20], have also found ethnic disparities in COVID-19 infection rates. ZIP codes with greater Black and Latinx populations had higher COVID-19 infection rates. Moreover, another study using ZIP code data for NYC found that for every ten percent point increase in Black residents and Latinx residents, there was an associated two percent increase in COVID-19 infection rates [21]. One additional study using data from cities and towns in Massachusetts found that during the initial months of the pandemic (through 6 May 2020), a 10% increase in the Black and Latinx population was associated with an increase of 312.3 and 258.2 COVID-19 cases per 1,000,000, respectively [22]. To the best of our knowledge, this study is the only study to have used statewide municipal-level data in the U.S. to examine ethnic disparities in COVID-19, with most research using the municipal unit of analysis conducted in other countries [20,23,24,25,26,27,28].

### 2.2. Causes of Ethnic Disparities in COVID-19 Infection

Studies examining factors that contribute to ethnic disparities in COVID-19 health outcomes in the U.S. have generally been conducted using individual-level data. These studies have found that socioeconomic inequalities increased vulnerability to COVID-19 infection and deaths, including factors such as living in poorer households, possessing a lower access to healthcare, job insecurity, living in places with a higher proportion of foreign-born residents, household size, living in places with a higher proportion of food service workers, and population density, and disproportionately affected Latinx and Black communities in the United States [9,22,29,30,31]. A systematic review evaluating the factors contributing to COVID-19 disparities across ethnic groups found that the primary drivers were differences in health care access and exposure risk, such as household crowding and population density [32]. Among Latinx people, various studies found that the main drivers were employment, housing, and immigration-related insecurity [29,30,33].

Few US longitudinal studies have been conducted that explore the consistency of vulnerability and exposure factors in terms of their association with ethnic disparities in COVID-19 infection over longer periods of time. One exception is a study that used individual-level, national data from June 2020 to July 2021 to assess the extent to which the social determinants of health contributed to racial/ethnic differences in COVID-19 infection rates [34]. This study considered the type of employment, household size, social distancing, level of education, and economic resources as potential contributors to COVID-19 infection rates. It found that Latinx people had an increased risk of COVID-19 infection but not Black respondents. However, none of the variables explained the elevated risk amongst the Latinx sample. Similarly, a study using data from towns in Massachusetts found that the association between the percentage of Black residents and COVID-19 cases decreased over time, while the association between Latinx residents and COVID-19 cases continued increasing across the five time periods [35]. Additionally, the percentage of essential workers had a significant positive association with COVID-19 infection across all time periods [35].

## 3. The Context of New Jersey

NJ, a state in the northeastern U.S. with approximately nine million residents, has unique characteristics that make it a theoretically important geographic setting for advancing the understanding of municipal-level ethnic disparities in COVID-19 case rates. First, NJ is by far the most densely populated state in the U.S. According to the 2020 U.S. Census [36], NJ had 1263.0 residents per square mile, compared to 1061.4 and 901.2 per square mile in Rhode Island and Massachusetts, which are the second- and third-most densely populated states, respectively. At the same time, NJ’s 565 municipalities differ from each other in terms of their land-use patterns. Urban developed lands are located in the northeastern, central, and western regions, in many cases adjacent to large cities including NYC and Philadelphia. However, there are also areas of largely undeveloped farmland and forests in the northwest and southeastern regions of the state.

NJ is also one of the most ethnically diverse states in the U.S. According to the 2020 U.S. Census, 21.6% of the population identified themselves as Latinx, and 12.4% as non-Latinx African American or Black. Since 2010, the Latinx population experienced the fastest decade-long increase in NJ, with the total proportion growing from 17.7% to 21.6%. In 2020, 23 out of 565 municipalities had a greater than 50% proportion of Latinx residents, and 7 had a greater than 50% proportion of African American or Black residents. While many municipalities with sizable Latinx or Black populations are in northeastern urban areas near NYC, some are scattered across the state and have formed clusters of Latinx and Black communities in local population centers across the southern and coastal regions of the state.

NJ’s municipalities are also highly segregated by ethnicity. For example, White students no longer represent the majority of NJ’s student population and only one in every five districts has student enrollment that is racially proportional to the county in which it is located [37]. Furthermore, NJ’s history of redlining—a practice of systematically denying financial support for homeownership in racially discriminatory ways—has further contributed to socio-spatial inequalities in wealth according to race throughout NJ [38,39].

In addition to the population density, ethnic composition, and segregation of NJ, the time course of the COVID-19 pandemic makes NJ’s municipal data an especially important resource for examining ethnic disparities and their causes over time. The first peak in case rates in NJ (during an initial wave of infections between March and June 2020), concentrated largely in the NYC metro area, was one of the early “hotspots” in the U.S. In fact, through the end of May 2020, NJ was the state with the second largest number of cases and deaths in the U.S. [40]. By mid-March 2020, there were sudden and mass shut-downs of businesses, schools, medical offices, and other aspects of social infrastructure [41]. At the same time, there was little guidance or systematic support regarding how to assess risk levels when engaging in daily life and how to adjust to new public health and socio-political landscapes. For example, it was not until the first week of April that the CDC released guidelines for wearing masks among the general public [42]. While the social upheaval caused by shutdowns and confusion about the best public health practices occurred across the country during early 2020, few other states had correspondingly high case rates during this first wave.

By the second peak (during a second wave of infections spanning from mid-October 2020 through mid-February 2021), case rates surged across all regions of NJ [43], and system-level responses—including public health guidance on social distancing and mask wearing—were broadly disseminated. The third peak (during the wave from mid-February 2021 to mid-May 2021) took place after the introduction of the COVID-19 vaccine in December of 2020 [41], but before the vaccine was readily available in local pharmacies and other community-based settings. Given the documented racial disparities in access to protective measures, e.g., regarding social distancing, testing sites, and vaccinations [44,45], especially during the initial year of the pandemic, the distinguishing characteristics of the societal context surrounding each of these three peaks suggest the importance of examining whether the extent of, and contributing factors to, disparities differ across multiple waves of infection.

## 4. Focus of the Current Study

This study examines differences in COVID-19 case rates across municipalities with relatively high percentages of Latinx and/or Black residents during three spikes in cases that occurred during the first 17 months of the pandemic in NJ. COVID-19 infections during this time were primarily caused by the original variant and the Alpha variant, as this period predated later surges in cases caused by the Delta and Omicron variants. Recognizing the unique context of NJ and the dearth of evidence on contributing factors to ethnic disparities in COVID-19 health outcomes at the municipal level, our study aims to examine the following hypotheses [H]: 

**H1.** 
*NJ municipalities with a relatively high proportion of Latinx residents and/or Black residents will have higher cumulative COVID-19 case rates compared to their counterparts across the first three peaks in infection rates.*


**H2.** 
*Municipal-level vulnerability and exposure factors will account for disparities differently across the three peaks in infection rates.*


Regarding H2, we draw on the World Health Organization’s (WHO) Health Inequity Causal Model to guide our examination of vulnerability and exposure factors [11]. This model considers various dimensions of social disparities that contribute to health inequities that span multiple social institutions, including the labor market, neighborhood infrastructure, healthcare, and families. Differential exposure factors are related to how social positions influence exposure to the virus leading to infection [11,46]. In this study, we consider household size, household and neighborhood crowding, occupation, distance from NYC, the use of shared transportation, neighborhood stability (i.e., the extent to which residents remain in their municipality over time), counts of long-term care beds, and political ideology as risk factors that can influence exposure to COVID-19 infection and potentially account for ethnic disparities in case rates. Differential vulnerability factors are those that describe how, after exposure occurs, one’s social position can make COVID-19 symptoms more severe for some individuals. Examples of such factors include socio-economic resources, environmental stressors, health status, and cumulative life course factors [11,46]. In this study, we selected municipal-level health status, older age, air pollution, English fluency, educational attainment, and income as differential vulnerability factors that can further explain the ethnic disparities in COVID-19 case rates.

While exposure and vulnerability factors are conceptually distinct, the data available to operationalize these factors have some overlap. We assigned variables to either vulnerability or exposure categories based on the prior literature and our own discussions of how and when these factors affect the exposure to and severity of COVID-19 infection. Furthermore, the exposure and vulnerability factors were calculated at the municipal level, which is consistent with findings from prior studies wherein both differential exposure and vulnerability factors were more prevalent in areas with higher proportions of Black and Latinx residents and were associated with a greater risk of COVID-19 infection and related death [20,23,41,46].

## 5. Materials and Methods

### 5.1. Data

In this paper, the dependent variable of interest is COVID-19 case rates at the municipal level. Since the state of NJ has not, as of the time of writing this article, publicly released municipal-level COVID-19 data, this paper used a dataset obtained by a research team that manually tracked daily releases of COVID-19 cases from county and municipal websites between 15 April 2022, and 15 May 2021. The data collection and cleaning methods are described extensively in a prior paper [43]. There are small differences between the data set in this paper and that described in the prior paper [43]: additional data became available, and subsequent analysis changed the dates corresponding to the three peaks in infection rates and their associated waves slightly. However, the collection and cleaning methods are identical.

Figure 1 portrays the trend of changes using new daily cases reported at the state level in NJ from 1 March 2020 to 31 August 2021. Wave 1 started upon the breakout of COVID-19 in NJ and reached a low point around 1 June 2020. Wave 2 lasted from 15 October 2020 to 10 February 2021. Wave 3 lasted from 10 February 2021 to 15 May 2021. Out of the total of 565 municipalities in NJ, 558 (98.8%) reported data for Wave 1, 406 (71.9%) reported data for Wave 2, and 371 (65.7%) reported data for Wave 3. Explanations for missing data are reported in the prior paper [43].

### 5.2. Measures

The COVID-19 municipal data, as described above, were integrated with sociodemographic data at the municipal level to create the dataset for our study. The data sources for measures of exposure and vulnerability factors are listed in Table A1.

#### 5.2.1. Dependent Variable: Cumulative Infection Rate

Cumulative infection rates were calculated as the cumulative cases during each of the three waves per 100k total population for municipalities. The total population size of each municipality for calculating the cumulative infection rate was extracted from the 2019 American Community Survey (ACS) five-year estimates.

#### 5.2.2. High percentages of Latinx and Black Municipalities

To create groups of municipalities based on ethnicity, we first ranked all municipalities in NJ by the percentage of Latinx residents. The upper quartile was labeled as municipalities with relatively high percentages of Latinx residents (RHL), and the rest as municipalities with a relatively low percentage of Latinx residents (RLL). We used the same approach to create municipalities with relatively higher percentage of Black residents (RHB) and municipalities with relatively lower percentage of Black residents (RLB). Since the distribution and percentage of non-Latinx Black residents in NJ differs from the distribution and percentage of Latinx residents, the quartile cutoff differs (8.6% for RHB; 15.7% for RHL). However, because cutoffs were defined relative to all municipalities in NJ, the thresholds and the classification of each municipality remained consistent throughout the three time points during the 17-month study period. Data for percentage of Latinx residents and percentage of non-Latinx Black residents were also from the 2019 American Community Survey (ACS) five-year estimates.

This quartile-based approach ensured sufficient sample size for each group. Moreover, this approach ensured that the RHB and RHL groups had equal numbers of municipalities. However, because the cutoff for inclusion into the “Relatively High” groups differed for Latinx and non-Latinx Black populations, we adopted a second classification strategy of using an identical cutoff of 20% or higher for both Latinx/non-Latinx Black residents (see “Analytic Strategy” below).

Among the total of 565 municipalities, 61.1% were RLL and RLB municipalities, 13.8% were RHL and RLB municipalities, 13.8% were RLL and RHB municipalities, and 11.3% were both RHL and RHB municipalities.

#### 5.2.3. Exposure and Vulnerability Factors

The complete list of exposure and vulnerability factors in this paper, as well as the measures used to operationalize those factors, are listed in Table A1. Risk factors were selected for inclusion in the model based on theoretical and methodological considerations. Previous literature on risk factors for COVID-19 motivated the theoretical constraints [22,23,26,27,32]. We then further narrowed our selection of risk factors based on the availability of municipality-level measures or those that could be constructed from census-tract level data. Some potential risk factors were excluded because of their high degree of collinearity with other risk factors, which caused methodological problems regarding the analytic strategy. For example, a municipality’s percentage of immigrants was removed from the model as it was highly correlated with the percentage of English spoken less than “very well” (*r* = 0.9). Following the WHO model (refer to Section 4), possible risk factors were then designated as either exposure or vulnerability factors.

Measures of exposure factors included household crowding, household size, percentage of residents employed in high-risk jobs, percentage of residents using shared transportation, population density, percentage of urbanized area within a municipality, point-to-point distance from the central point of each municipality to NYC, stability in the municipality, the number of long-term care beds in a municipality, and average political ideology in the municipality. For example, the exposure factor of stability, measured as percentage of a municipality’s residents who lived at the same address year-over-year, may lead to lower risks, as frequent moves may lead to the transmission of the virus. As a second example, NYC was the local center of the pandemic, especially during the first peak in infections. A municipality’s proximity to NYC may increase the likelihood that residents would commute to NYC and thus increase their likelihood of exposure to the virus. Political ideology was included as an exposure factor because literature suggests that political ideology predicts the adoption of preventive strategies against infections [47,48].

Measures of vulnerability factors included health status, percentage of the municipality’s population who were older adults, access to acute care facilities, income inequality, language barriers, poverty, education attainment, unemployment rate, and air pollution. For example, access to acute care facilities may predict testing and isolation, which can improve outcomes at the municipal level. Long-time exposure to air pollution, measured as the concentration of fine inhalable particles in the air that measure less than 2.5 micrometers in diameter (PM 2.5), increases susceptibility to COVID-19 [49].

Since this analysis aims to explain differences in COVID-19 infections rates across municipalities, all measures of risk were included at the municipal level, rather than the individual level. Most of the measures were available at the municipal level. Exceptions included percentage of urbanized areas and distance to NYC, which were calculated from map using QGIS Field Calculator and Distance Matrix functions. Point data were collected for locations of long-term care facilities and acute care facilities. Concentration of PM 2.5 was originally reported at census tract-level. We aggregated point data and census tract data at the municipality-level. Appendix A Table A1 provides further explanation about each measure.

#### 5.2.4. Covariates

In NJ, Latinx and Black residents, in many cases, co-reside in the same municipalities. Therefore, we controlled for percentage of Latinx residents in models contrasting RHB to RLB municipalities. The inclusion of this covariate allows for stronger inferences concerning the infection rate gaps for the focal ethnic groups. Similarly, we included percentage of Black residents in the models for RHL/RLL municipalities. We further included a covariate regarding percentage of Asian residents in all models.

### 5.3. Analytic Strategy

To test H1, we first used basic inferential statistics (an independent samples *t*-test, two-tailed) to test whether infection rates in RHL and RLL municipalities differed, as well as between RHB and RLB municipalities. We refer to the average difference between infection rates by municipal ethnic composition as the *infection rate gap*. We tested for infection gaps in each of the three waves illustrated in Figure 2 separately.

We then tested H2 by exploring the potential factors that contributed to the gaps across the three waves. First, we examined correlations of potential exposure and vulnerability factors with infection rates at each of the three waves. Pairwise deletion was used to treat missing data from the correlation analysis. Then, we conducted independent *t*-tests to examine whether there were significant differences in exposure and vulnerability factors by municipal ethnic composition across each wave. Lastly, we performed the Oaxaca–Blinder decomposition analysis [12,13] to test the extent to which the exposure and vulnerability factors contributed to the infection rate gaps at each time point. This analytic approach was conducted to contrast RHL to RLL municipalities, and then replicated to contrast RHB to RLB municipalities.

The Oaxaca–Blinder decomposition method was initially developed to examine gender and racial discrimination in wages. The method can decompose the gap between two groups into explained and unexplained parts [50,51]. Taking the contrast of RHL to RLL municipalities as an example, we first estimated two separate linear regression models with *x*_1_ to *x_k_* variables (exposure, vulnerability factors, and covariates) predicting COVID-19 case rates (*Y*) for RHL and RLL municipalities. b represents the estimated coefficient for each predictor. Adapted from [51], the gap in COVID-19 case rates between the two group can be expressed as:$${\bar{Y}}^{RHL}-{\bar{Y}}^{RLL}=\overset{k}{\underset{j=1}{\sum }}b_{j}^{RHL}{\bar{x}}_{j}^{RHL}-\overset{k}{\underset{j=1}{\sum }}b_{j}^{RLL}{\bar{x}}_{j}^{RLL}+b_{0}^{RHL}-b_{0}^{RLL}$$

The equations can be decomposed into two parts, with the explained part attributed to the difference in predictors between RHL and RLL municipalities (${\bar{x}}_{j}^{RHL}-{\bar{x}}_{j}^{RLL}$), and the unexplained part attributed to differences in coefficients and unobserved variables [51]. Differences in *b^RHL^* and *b^RLL^* among the non-discriminating coefficients (*b^ALL^*, estimated in a pooled model) are usually considered to be due to discrimination [51].
$${\bar{Y}}^{RHL}-{\bar{Y}}^{RLL}=\overset{k}{\underset{j=1}{\sum }}b_{j}^{ALL}({\bar{x}}_{j}^{RHL}-{\bar{x}}_{j}^{RLL})+\left[\overset{k}{\underset{j=1}{\sum }}{\bar{x}}_{j}^{RHL}\left(b_{j}^{RHL}-b_{j}^{ALL}\right)+\overset{k}{\underset{j=1}{\sum }}{\bar{x}}_{j}^{RLL}\left(b_{j}^{ALL}-b_{j}^{RLL}\right)\right]+(b_{0}^{RHL}-b_{0}^{RLL})\;\;\mathrm{Explained part}\;\;\;\;\;\;\;\;\;\;\;\;\;\mathrm{Unexplained part}$$

The Oaxaca–Blinder decomposition method has been used to examine drivers of COVID-19-related inequality across racial/ethnic groups, including hospitalizations, all-cause mortality, and unemployment at individual-level [52,53]. However, no study has used the method to examine explanatory factors for spatial inequality of COVID-19 infections at the municipal level, even though regional-level characteristics have been widely examined as key factors driving the spread of COVID-19 in the US [16,21].

Decomposition models were run for both the quartile and 20%-or-more-of-the-population cutoffs. For each wave, missing data were handled using listwise deletion. Since the number of missing data points increased with each successive wave, the decomposition analysis was run with successively fewer data points. Additional sensitivity tests using samples with only municipalities that reported data cross all three time points were conducted to ensure that results were not an artifact of changing sample size.

Given the spatial nature of the dataset, we tested spatial autocorrelation of residuals of the above-mentioned linear regression models. However, due to the presence of a large proportion of missing data in Waves 2 and 3, the use of spatial models may be inappropriate because they may yield additional bias [54]. Therefore, we incorporated spatially lagged dependent variables to estimate the decomposition models as another set of sensitivity tests. The results were robust across almost all predictors.

## 6. Results

### 6.1. The Distribution and Trend of COVID-19 Infections in NJ

The infection rates varied widely across the municipalities in each of the three waves shown in Figure 1A. Compared to Waves 2 and 3, the infections in Wave 1 were confined to a much narrower range of municipalities. During Wave 1, five municipalities reported 0 infections, and an additional 85 reported infection rates of less than 500 per 100k, whereas during Waves 2 and 3, the infection rates were distributed across more municipalities. Every municipality reported more than 500 per 100k infection rates by Wave 2. The average cumulative infection rates were 1331 per 100k in Wave 1, 4443 per 100k in Wave 2, and 2571 per 100k in Wave 3. This variety of infection rates between waves illustrates the need for an explanatory analysis for each wave. Frequency histograms showing the distribution of cumulative infections rates for the three waves are available in Appendix A (Figure A1).

The maps in Figure 2 also demonstrate how the distribution of infection rates changed over the three time points. The infections were concentrated in the greater NYC metro area in Wave 1. This area largely surrounds the commuter lines in and out of Manhattan. In Wave 2, while areas near NYC continued to suffer from high infection rates, there were also high rates of infection near the Philadelphia metro region and in the more rural farmland in southern NJ. In Wave 3, high infection rates were observed across the state.

### 6.2. Correlates of Infection Rates with Exposure and Vulnerability Factors

The correlation matrix in Table 1 indicates that the majority of the exposure and vulnerability factors were correlated with infection rates in at least one wave. Regarding the exposure factors, the five factors with the strongest (and statistically significant correlates) of infection rates in Wave 1 were household crowding, distance to NYC, household size, long-term care beds, and population density. With respect to vulnerability factors, the five strongest correlates of infection rates in Wave 1 were the percentage of residents having less than a high school degree, the all-cause mortality rate, the percentage of residents reporting that they speak English less than “very well”, per capita income, and counts of acute care facilities. Among the abovementioned exposure and vulnerability factors, all except distance to NYC and per capita income had positive correlations with infection rates in Wave 1. Residing farther away from NYC was associated with lower infection rates in Wave 1, as was higher income.

In Wave 2, fewer exposure factors were significantly correlated with infection rates. Notably, a municipality’s distance to NYC was no longer correlated with its infection rate. However, population density and household crowding remained significantly correlated, and the percentage of residents working in the transportation or food industries were also significantly correlated with infection rates. In Wave 2, the vulnerability factor of all-cause mortality was no longer strongly correlated with infection rates, but correlations between the percentage of a municipality’s residents having less than a high school degree or speaking English less than “very well” and per capita income remained strongly correlated.

In Wave 3, the exposure factors correlated with infection rates again included distance to NYC and the percentage of residents working in transportation. The vulnerability factors correlated with infection rates in Wave 3 were similar to those in Wave 2.

### 6.3. Gaps between RHL and RLL Municipalities

To test H1, we performed *t*-tests between infection rates in RHL and RLL municipalities in each of the three time periods (Table A2). In all three waves, RHL municipalities reported significantly higher infection rates than RLL municipalities (*p* ≤ 0.001). The infection rate gap, defined as the difference in infection rates between RHL and RLL municipalities, was smaller in Wave 3 (599.3, *t* = 5.94) compared to Wave 1 (1108.9, *t* = 9.12) and Wave 2 (1320.0, *t* = 8.85). To provide some context, in Wave 1, the average cumulative infection rate in RHL municipalities was more than twice as high as the average infection rate in RLL municipalities (data from Table A2; RHL = 2166/100K; RLL = 1057/100K; ratio of RHL to RHL = 2.05).

The maps in Figure 2 provide additional visualization of the gap between RHL and RLL municipalities. In Figure 2, the shaded areas representing RHL municipalities tended to have higher infection rates than other areas. Further visualization is provided by the histograms in Appendix A Figure A2, demonstrating the frequency distributions of infection rates contrasting RHL with RLL municipalities.

To test H2, we examined how the exposure and vulnerability factors explained the RHL/RLL gap in each of the three time periods. To do so, we first determined how the vulnerability and exposure factors themselves differed in RHL/RLL communities. Second, we determined which of these differences accounted for the RHL/RLL gap in each of the three pandemic waves.

The RHL and RLL communities exhibited differences in many exposure and vulnerability factors (Table A2). Compared to RLL municipalities, RHL municipalities were geographically closer to NYC and had significantly higher percentages in terms of non-Latinx Black residents, crowded households, larger household sizes, food industry and transportation workers, workers carpooling or using public transportation, population density, urbanized areas, worse air pollution, acute care facilities, populations experiencing a language barrier, populations with a less than high school degree, and greater unemployment rates. In addition, RHL municipalities were found to have lower percentages in terms of utilities workers, all-cause mortality rates, older adults, Trump voters, lower per capital income, and lower neighborhood stability.

Next, we employed a decomposition analysis (see Section 5.3) to determine which of these exposure and vulnerability factors explained the RHL/RLL gap at each time point. The results are reported in Table 2 solely for statistically significant factors. Broadly, the exposure and vulnerability factors tested explained the majority of the RHL/RLL infection rate gap across all three waves (Wave 1: 63.3%; Wave 2: 62.7%; Wave 3 58.8%), but the relative importance of the factors changed across the three time points.

In Wave 1, for example, the exposure factors of the distance to NYC and household size explained 21.9% and 9.3% of the RHL/RLL gap, respectively, while the vulnerability factor of education, as measured by the percentage of a municipality’s residents with an at least high school education, explained 53% of the RHL/RLL gap. Interestingly, since RHL municipalities had lower all-cause mortality than RLL municipalities, this vulnerability factor actually made the gap smaller than it would have been otherwise, specifically, by 353.6 cases per 100K (a reduction of 32%).

In Wave 2, the exposure factors of the distance to NYC and household size no longer explained the RHL/RLL gap. Instead, the exposure factors of household crowding, percentage of residents who worked in transportation, and percentage of urbanized areas explained 24.4%, 20.0%, and 6.9% of the gap, respectively. One additional exposure factor, political ideology as measured by the percentage of Trump voters, also explained part of the gap in Wave 2. RHL municipalities had lower percentages of Trump voters than RLL municipalities (38% vs. 48%; see Appendix A Table A2), and this difference contributed to reducing the gap by 215.4 cases per 100K (a reduction of 16.3%). In Wave 2, as in Wave 1, the vulnerability factor of all-cause mortality, lower in RHL than in RLL municipalities, also narrowed the gap by 91.4 cases per 100K. In Wave 2, one new vulnerability factor emerged. Language barriers, measured as the percentage of residents speaking English less than “very well”, explained 49.5% of the gap.

In Wave 3, as in Wave 1, the exposure factor of distance to NYC explained 50.3% of the gap in infection rates between RHL and RLL municipalities. Unlike Waves 1 and 2, neither household crowding nor household size contributed significantly to the gap. However, the percentage of transportation workers explained a percentage of the gap similar to that in Wave 2 (19.5%). As in Waves 1 and 2, the vulnerability factor of all-cause mortality rate remained a significant factor that narrowed the gap by 44.6 cases per 100k.

To confirm that the results of the decomposition analysis were not unique to the particular method for classifying municipalities into RHL and RLL groups, we repeated the decomposition analysis with the 20%-or-higher-of-the-population cutoff (see Section 5.2.2). This analysis (Table A3) confirmed that all patterns remained stable when using this second classification method. Furthermore, the sensitivity analysis regarding only the municipalities that reported data across all three time periods indicated that the majority of the results were robust while some became marginally significant with a smaller sample (Table 2). The models wherein the spatially lagged dependent variable was added as a predictor also showed robust findings for the majority of the factors (Table 2).

### 6.4. Gaps between RHB and RLB Municipalities

The statistical tests supported H1 for RHB/RLB communities as well, revealing a significant gap in the infection rates between RHB and RLB municipalities in Waves 1 and 2, but not in Wave 3. The gap was 303.6 per 100k in Wave 1 (*t* = 2.36, *p* ≤ 0.05) and 347.4 per 100k in Wave 2 (*t* = 2.17, *p* ≤ 0.05). In contrast, in Wave 3, the RHB municipalities reported lower infection rates, on average, than RLB municipalities, though this difference was not significant (*t* = −1.81, *p* = 0.07). Further visual context is provided by histograms of infection rates in RHB and RLB municipalities (Appendix A Figure A3). Although statistical tests revealed an RHB/RLB gap, the maps in Figure 2 do not show visually salient differences between RHB municipalities and RLB municipalities.

As we did for the RHL/RLL gap, we tested H2 for the RHB/RBL gap by first determining which exposure and vulnerability factors were different in the RHB and RLB municipalities, and then using the decomposition analysis to see which factors explained the gap across each of the three time points.

On average, the RHB and RLB municipalities also showed differences regarding their exposure and vulnerability factors (Table A4). Compared to RLB municipalities, RHB municipalities were farther away from NYC and had significantly higher percentages in terms of Latinx residents; crowded households; household size; healthcare workers; workers in the public safety, food, and transportation industries; populations using carpool or public transportation; population density; urbanized areas; worse air pollution; long-term care beds; acute care facilities; populations with language barrier; populations with less than high school degree; and unemployment rates. In addition, RHB municipalities had lower percentages in terms of older adults, Trump voters, per capita income, and lower neighborhood stability.

The findings from the decomposition analysis contrasting RHB with RLB municipalities, as displayed in Table 3, indicated that the total proportion of the gap explained by all the included factors was only significant in Wave 1 (83.81%). (We do not present the analysis for Wave 3 since the prior analysis did not demonstrate a statistically robust infection rate gap between RHB and RLB communities at that period.) However, because some factors increased the gap and others decreased the gap, it is still possible for individual factors to show significant contributions to the infection rate gap even if the overall decomposition model does not significantly explain the infection rate gap.

In Wave 1, the exposure factor of the distance to NYC narrowed the gap by 152.2 cases per 100k. Unlike the RHL municipalities, the RHB municipalities had relatively longer distances to NYC than RLB municipalities. In Wave 2, the exposure factors of a municipality’s percentage of transportation workers and household crowding explained 58.6% and 45.8% of the gap. RHB municipalities had lower percentages of residents voting for Trump, which narrowed the gap by 497.9 cases per 100k. The factors that explained the RHB/RLB gap in Wave 2 were similar to the Wave 1 factors in the RHL/RLL model, except that as the center of Wave 2 was no longer NYC, the relatively longer distance to NYC of the RHB municipalities enlarged the gap by 124.8 cases per 100k.

We repeated the decomposition analysis using the 20%-or-higher-of-the-population cutoff as a sensitivity analysis (Table A3). Two minor changes were observed. First, the distance to NYC was no longer a factor during either of the waves. Second, the unemployment rate was found to narrow the RHB/RLL gap in Wave 1. The sensitivity analysis using only municipalities that reported data across all three waves reported robust results for all the exposure and vulnerability factors (Table 3). The models wherein the spatially lagged dependent variable was added as a predictor yielded robust findings for the majority of the factors (Table 3).

## 7. Discussion

This study provides empirical evidence of ethnic disparities regarding COVID-19 at the municipal level in NJ [H1]. It also provides insight into the exposure and vulnerability factors that may underlie these disparities and how they changed across three peaks in infection rates during the first 17 months of the pandemic [H2].

Consistent with H1, both RHL and RHB municipalities reported a significant infection rate gap relative to their respective counterparts (RLB and RLL municipalities). RHL and RHB municipalities had significantly higher COVID-19 infection rates than RLL and RLB municipalities. The RHL/RLL gap was significant across all three waves. In contrast, the RHB/RLB gap was significant across Waves 1 and 2, but not Wave 3.

Moreover, as guided by the WHO Inequality Casual Model, we identified both key exposure and vulnerability factors that differed between RHL and RHB municipalities and could, therefore, potentially account for the ethnic disparities as quantified by the infection rate gap. Consistent with H2, we found that these risk factors explained a majority (~60%) of the infection rate gap across all three time points, and that the contribution of specific factors changed over the course of the pandemic.

At various times over the first 17 months of the pandemic, the exposure factors that contributed significantly to the observed ethnic disparities included being closer to NYC, a larger household size, household crowding, a higher percentage of transportation workers, and a higher percentage of urbanized areas. The vulnerability factors that contributed significantly to the observed ethnic disparities included language barriers and education. In contrast, two risk factors served to decrease the size of the observed ethnic disparities: the exposure factor of political ideology in both RHL and RHB municipalities, and the vulnerability factor of health status in RHL municipalities.

In the remainder of the discussion, we provide context to help interpret why these specific exposure and vulnerability factors accounted for the observed ethnic disparities and why the relative contributions of these factors might have changed over the first 17 months of the pandemic. We also address the empirically observed similarities and differences between the RHL/RLL and RHB/RLB infection gaps and the exposure and vulnerability factors that explain them.

The result wherein the distance to NYC contributed to the overall infection rates is unsurprising. Prior studies conducted with county-level data in NJ similarly found that the distance to NYC was a primary explanatory factor for COVID-19 infection rates [52]. In NJ, the municipalities closest to NYC also have the largest number of Latinx residents in the state [36]. These municipalities are in Hudson County and are some of the most densely populated municipalities in the state [36]. For example, two of these municipalities, Union City and West New York, are among the most densely populated municipalities in the world, and both have a Latinx population over 75% [36]. In this sense, Latinx communities were on the geographic frontlines of the pandemic for the state of NJ, with increased exposure to contracting COVID-19 as it spread across the greater NYC metro area. In contrast to RHL municipalities, RHB municipalities tended to be farther from New York City. For example, four of the ten municipalities with the highest percentages of Black residents are in southern New Jersey, far from New Yok City (Lawnside, in Camden County; Willingboro, in Burlington County; Salem City, in Salem County; and Fairfield Township, in Cumberland County). This average difference between RHB and RHL municipalities can explain why the exposure factor of distance to NYC was associated with widening the RHL/RLL infection rate gap and narrowing the RHB/RLB infection rate gap during Wave 1.

Interestingly, the exposure factor of the distance to NYC significantly predicted the RHL/RLL infection rate gap in Waves 1 and 3, but not in Wave 2. An inspection of Figure 1 shows that during Waves 1 and 3, the infection rates were high in NJ but relatively low in the country as a whole. In contrast, during Wave 2, the infection rates were high both in NJ and in the country as a whole in Wave 2. We speculate that discrepancies between NJ’s infection rates and the country’s infection rates reflect the introduction of new variants, which may spread earlier through large cities that have strong connections to global travel. Further studies could address whether similar patterns emerge near other cities that are closely connected to global travel.

Several of the exposure factors that predicted infection rate gaps were associated with living and working conditions. Various scholars have identified that poverty and economic factors such as household size and crowding, urbanization, and working in transportation and other essential services are primary drivers of COVID-19 infection in historically oppressed communities [55,56,57]. In NJ, the Latinx population accounted for 22% of all front-line workers, with 54% working in building-cleaning services; 38% working in trucking, warehouse, and postal services; and 22% in transportation services [53]. Overall, this population is overrepresented in low-wage and high-risk occupations [58] increasing their risk of exposure to COVID-19. Prior studies have discussed that an increased risk of exposure coupled with a lower capacity for social distancing are among the primary contributors to greater COVID-19 infection rates among Latinx and Black communities [59,60]. In regions that are ethnically segregated, as is the case across many municipalities in NJ, this leads to a concentration of social disadvantages that places a greater number of residents at risk of infection and limits their capacity to prevent the spread given the density of the environments in which they reside [60]. In NJ, racial segregation has placed Latinx communities at greater risk given their dense concentration in small geographic areas where household crowding and urbanization is common—two exposure factors that were found to be significant contributors to municipal-level ethnic disparities in COVID-19 infection throughout the other time points in our analysis. This further lends evidence to the negative health consequences of systemic racial and economic inequality that have been documented in the prior literature and that have contributed to ethnic disparities in COVID-19 infection rates in NJ and across the United States [61].

The contributions of these exposure factors related to living and working varied across waves. During Wave 1 (i.e., from March of 2020 through 1 June 2020), larger household sizes explained part of the infection rate gap (for RHL municipalities), but in Wave 2, (i.e., 15 October 2020, to 10 February 2021), household crowding emerged as a robust explanatory factor (for both RHL and RHB municipalities). This pattern of findings may reflect shifts in the general knowledge regarding the protective measures for lowering the risk of infection, alongside structural conditions that influence one’s ability to act on such knowledge (for further discussion, see Garcia et al. [62] and Ray [63]). For example, in Wave 1, there was less understanding of viral transmission, meaning that the exposure factor of greater household size, which was more prevalent among RHL municipalities, contributed to the infection rate gap. However, during Wave 2 there was more widespread understanding of household transmission and more readily available mitigation measures such as masking and physical distancing that might protect those with a large household size but sufficient space. This knowledge would not change the impacts of household crowding, where presumably even masking was ineffective and isolation was impossible. This is an example of how structural conditions can prevent people from being able to act on knowledge regarding effective social distancing and other mitigation measures [64].

This pattern of better available information coupled with an (in)ability to act on it may also explain the pattern across the waves of vulnerability factors such as education and language barriers. For example, in RHL municipalities, a lower level of education was a significant vulnerability factor in Wave 1, but not in Wave 2. In contrast, English proficiency was a significant vulnerability factor in Wave 2, but not in Wave 1. In Wave 1, a lower level of education might have contributed to limited health literacy regarding infectious disease generally, especially when public health knowledge of COVID-19 was not yet broadly developed and disseminated [65,66]. By Wave 2, at which time public health information was more available, residents with limited English proficiency would have been less likely to benefit [67,68], thus contributing to ethnic disparities. Previous research has established that limited English fluency is a risk factor for COVID-19 infections, possibly because Latinx communities have reported difficulties in accessing reliable translated health information during the pandemic [67]. In a state where over one fifth of the population is Latinx and 2.6 million do not speak English at home [36], the need for translated health and safety information is clear.

Another change over time was observed in the relationship of political ideology to infection rate gaps. During Waves 2 and 3, but not during Wave 1, we found evidence that political ideology reduced the RHL/RLL and RHB/RLB infection rate gaps. Over time, the response to the pandemic became an increasingly political issue. Partisanship led to different attitudes towards adopting preventive strategies against COVID-19 transmission, with supporters of Trump less likely to engage in mitigation strategies such as masking and physical distancing. [47,48,69]. Although NJ showed relatively low support for Trump overall compared to other parts of the US, there was considerable variation between municipalities. On average, RHL and RHB municipalities in NJ showed lower support for Trump than their RLL and RLB counterparts (RHL = 38%, RLL = 48%; RHB = 35%, RLB = 49%). Thus, the exposure factor of political ideology in this case was associated with a reduction in the infection rate gap.

Finally, by Wave 3, we found more limited evidence of exposures and vulnerability factors contributing to ethnic disparities. By Wave 3, there was no infection rate gap between RHB and RLB municipalities, and the size of the infection rate gap between RHL and RLL municipalities was smaller than in Waves 1 and 2. It is important to note that in Wave 3 (i.e., 10 February to 15 May 2021), COVID-19 vaccinations started to become more widely available. In general, studies have found lower vaccination rates for Latinx and Black groups relative to White and Asian groups [45,70]. Nevertheless, our study did not find an increased disparity in municipal-level infection rates between RHL and RHB municipalities and their counterparts (RLL and RLB). It is possible that the higher infection rates during Wave 1 and Wave 2 for RHL and RHB municipalities lead to higher levels of natural immunity because of prior exposure, thereby reducing disparities between high and low municipalities in Wave 3. Overall, these findings suggest the importance of continued attention among policymakers, practitioners, and researchers alike to how shifting macro-social contexts during the COVID-19 pandemic and other public health emergencies influence the manifestation and exacerbation of population health disparities.

There are several possible explanations for why our study found a larger and more robust infection rate gap related to the percentage of Latinx resident in municipalities relative to the results for municipalities with relatively high percentages of Black residents. First, it is possible that in NJ, the exposure and vulnerability factors in RHL and RHB municipalities differ from each other in significant ways, and that these differences are more salient in NJ than in other parts of the United States. As previously discussed, for example, RHL municipalities in NJ tended on average to be closer to NYC than RHB municipalities. Since proximity to NYC was a strong predictor of infections, RHL municipalities had an elevated risk compared to RHB municipalities. In addition, while our sample did not include data on individuals with an undocumented status, NJ is home to 440,000 undocumented immigrants, the fifth largest population in the country, of which 71% are Latinx [71]. Some scholars have discussed this population’s disproportionate risk for COVID-19 infection during the first two years of the pandemic when they were not eligible for federal economic relief and had to continue working [72,73].

A second possibility is that the relative disparity is an artifact resulting from the characteristics of the dataset or analytic methods rather than a true difference in outcomes between RHL and RHB municipalities. As described in the methods, not all municipalities reported data, and the number of municipalities with missing data increased over the pandemic. Thus, selection bias could have biased the pattern of findings. In addition, our primary technique for separating municipalities into relatively high and relatively low groups relied on rank rather than an absolute threshold. This means that, on average, the percentage of Latinx residents RHL municipalities was higher than the percentage of Black residents in RHB municipalities. However, although an artifactual explanation is possible, several sensitivity analyses suggest it is unlikely. Even when the analysis was run solely on the municipalities that reported data for all three waves, the pattern of results was similar. In addition, assigning municipalities to RHL and RHB groups based on absolute percentages (>20%) rather than relative rankings also showed similar patterns. Additional research drawing on more complete datasets within NJ (when available) and data in other states is necessary for more a more comprehensive interpretation of our finding that the COVID-19 disparity was larger for relatively high Latinx municipalities than for relatively high Black municipalities and municipalities with relatively low Black and Latinx residents.

## 8. Limitations

It is important to note the key study limitations related to the availability and quality of the data. First, the data allowed for the examination of only one COVID-19 population health outcome: infection rates. We were not able to examine other important COVID-19 outcomes, such as death rates, hospitalizations, or the economic repercussions of COVID-19, which might demonstrate different patterns of municipal-level disparities and explanatory factors over time. Second, the missing data constitute another study limitation. Due to the methods of data collection and changes to public reporting, the number of municipalities with valid data decreased over time. Although the sensitivity analyses showed similar patterns of results with the subset of municipalities that reported data across all three waves, attrition of any sort leaves open the possibility for selection bias. Third, the data only allowed for the examination of aggregation at the municipal level, while many exposure and vulnerability factors are likely to function at both the individual and municipal levels. Distinguishing between community-level or environmental effects and aggregated individual-level effects is critical for understanding the impacts of COVID-19 and policy responses. Fourth, our analysis of how contributors to ethnic disparities in COVID-19 changed over distinct time points was limited to the first 17 months of the pandemic in NJ during which the Alpha variant was predominant. The use of datasets that incorporate additional time points when other variants were more predominant is an important direction for future studies. Finally, some potential factors were not included due to data unavailability, such as the data-tracking mobility of individuals aggregated at the municipal-level and more detailed measures for municipal-level health conditions.

## 9. Conclusions

Despite the above limitations, this study contributes additional empirical evidence of ethnic disparities regarding COVID-19 infection by examining the municipal-level rates in one of the most densely populated, socioculturally diverse, and spatially segregated states in the US. The findings indicate the importance of continuing to research the growing health disparities between communities with relatively high percentages of Latinx and Black residents and communities with lower representations from these groups. In addition, there is also a need to attend to the ways in which the risk factors accounting for the disparities changed throughout the pandemic. Continued research on socio-spatial disparities in public health emergencies, especially in the context of systemic and structural racism, is essential for optimizing policy responses. Standard epidemiological models typically assume that all individuals and geographic areas have an equal risk of exposure and vulnerability, and most geospatial research studies on COVID-19 draw on data at the county level. As our findings indicate, studies that employ a more nuanced empirical characterization of risk and resilience at the municipal level have strong potential to provide better information to state and regional policymakers seeking to address health equity across ethnic groups.

## Figures and Tables

**Figure 1 ijerph-19-13963-f001:**
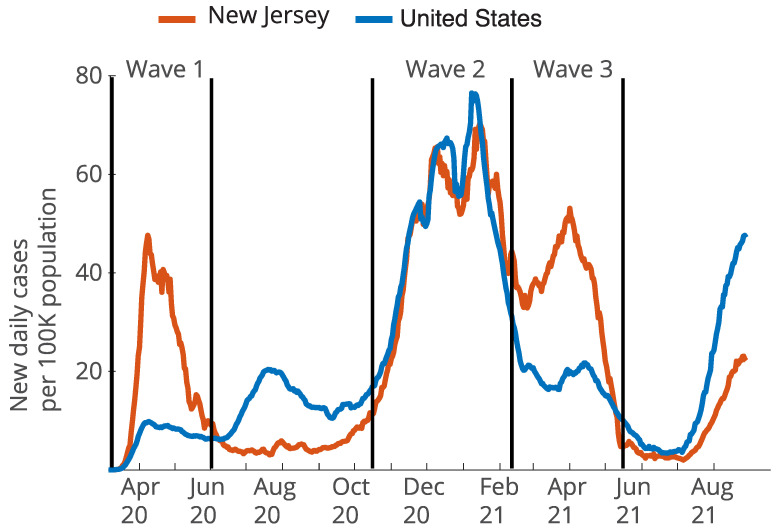
New daily COVID-19 cases per 100k population in New Jersey and the United States. Notes: Data were obtained from the COVID-19 Data Repository by the Center for Systems Science and Engineering (CSSE) at Johns Hopkins University. The plot was created from the raw data files by dividing new daily cases by population and then smoothing over a 7-day window.

**Figure 2 ijerph-19-13963-f002:**
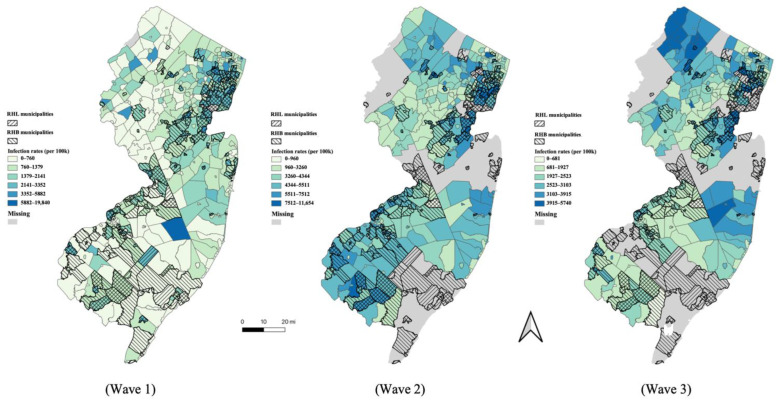
Distributions of COVID-19 within New Jersey municipalities characterized as Relatively High Percentages of Latinx Residents (RHL) and Relatively High Percentages of Black Residents (RHB) across three peaks in infection rates during the first 17 months of the pandemic.

**Table 1 ijerph-19-13963-t001:** Correlations of potential exposure/vulnerability factors with infection rates.

	Infection Rates Wave 1	Infection Rates Wave 2	Infection Rates Wave 3
Infection rates Wave 2	0.312 ***		
Infection rates Wave 3	0.303***	0.481 ***	
**Exposure factors**			
Household crowding	0.360 ***	0.396 ***	0.138 **
Household size	0.288 ***	0.078	0.018
% Healthcare workers	0.060	0.071	−0.082
% Public safety workers	−0.077	0.136	−0.030
% Food industry workers	0.015	0.181***	0.091
% Transportation workers	0.223 ***	0.450 ***	0.250 ***
% Utilities workers	−0.132 **	0.011	−0.093
% Carpool or public transportation	0.258 ***	0.143 **	−0.020
Population density	0.270 ***	0.223 ***	0.155 **
% Urbanized areas	0.160 ***	0.100 *	0.005
Distance to NYC	−0.314 ***	0.002	−0.288 ***
Stability	−0.007	0.001	0.123 *
Long-term care beds	0.272 ***	0.145 **	0.035
% Trump voters	−0.161 ***	0.008	0.155 **
**Vulnerability factors**			
All-cause mortality rate	0.390 ***	0.095	0.102 *
% 65+	−0.052	−0.115 *	0.001
Acute care facilities	0.157 ***	0.175 ***	0.030
Gini	0.100 *	−0.120 *	−0.048
% English less than “very well”	0.430 ***	0.383 ***	0.233 ***
Per capita income	−0.188 ***	−0.386 ***	−0.189 ***
% Less than high school	0.401 ***	0.429 ***	0.173 ***
% Unemployed	−0.013	0.080	0.007
Air pollution (PM 2.5)	0.072	−0.004	−0.189 ***
**Covariates**			
% Latinx residents	0.460 ***	0.518 ***	0.313 ***
% non-Latinx Black residents	0.168 ***	0.089	−0.057
% non-Latinx Asian residents	0.075	−0.109 *	−0.102 *

Notes. * *p* < 0.05, ** *p* < 0.01, and *** *p* < 0.001. Pairwise deletion was used to estimate the correlations. Appendix A Table A1 relates the factor labels in Column 1 to the variables they measure and the data sources.

**Table 2 ijerph-19-13963-t002:** Decomposition analysis contrasting RHL with RLL municipalities.

	Diff (RHL-RLL) ^a^	Wave 1 (*n* = 558)			Wave 2 (*n* = 406)			Wave 3 (*n* = 371)		
		B	SE	% Diff	B	SE	% Diff	B	SE	% Diff
Infection rates (RHL)		2166.3	91.4		5434.8	162.2		3020.5	92.3	
Infection rates (RLL)		1057.3	62.9		4114.8	66.9		2421.2	49.5	
Difference		1108.9 ***	110.9	100.0	1320.0 ***	175.5	100.0	599.3 ***	104.7	100.0
**Explained**		702.0 ***	109.5	63.3	828.3 ***	204.1	62.8	352.3 ***	103.2	58.8
**Exposure factors**										
Household crowding	3.04 ***	80.9	82.4		322.5 *	133.5	24.4	−8.6	68.3	
Household size	0.13 ***	102.9 **	34.2	9.3	−43.5	29.4		−39.7	26.5	
% Healthcare workers		−1.9	5.7		−32.0	25.1		−3.5	7.1	
% Public safety workers		4.5	7.0		−9.5	11.8		8.1	8.8	
% Food industry workers		−29.2	20.2		6.3	30.0		0.6	22.7	
% Transportation workers	1.76 ***	4.3	37.6		264.5 ***	73.8	20.0	117.0 *	49.9	19.5
% Utilities workers		17.5	12.4		6.2	19.0		7.7	13.2	
% Carpool or public transportation		−9.2	62.0		204.4	131.9		−107.4	55.5	
Population density		62.8	56.0		−139.5	92.2		5.8	64.3	
% Urbanized areas	17.65 ***	−21.6	23.9		91.2 *	37.4	6.9	−39.5	28.6	
Distance to NYC	−16.18 ***	242.7 ***	54.6	21.9	−105.8	71.2		301.6 ***	69.3	50.3
Stability		−5.8	14.7		−34.0	34.2		−37.9	23.8	
Long-term care beds		13.4	11.1		10.5	15.3		0.7	2.5	
% Trump voters	−10.04 ***	−27.0	51.3		−215.4 *	101.3	−16.3	−105.7	69.9	
**Vulnerability factors**										
All-cause mortality rate	−185.28 ***	−353.6 ***	87.3	−31.9	−91.4 *	36.7	−6.9	−44.6 *	19.2	−7.4
% 65+		35.1	42.1		116.9	74.5		68.6	49.4	
Acute care facilities		−22.2	16.1		22.1	27.3		5.6	13.6	
Gini		−10.5	14.9		−5.1	22.7		−1.6	8.7	
% English less than “very well”	12.41 ***	134.0	208.1		653.9**	231.5	49.5	111.6	117.1	
Per capita income		−125.3	78.0		−165.1	164.5		172.5	138.6	
% Less than high school	7.88 ***	591.0 *	276.1	53.3	91.2	158.6		−17.6	97.9	
% Unemployed		−27.2	16.9		−19.2	18.2		5.4	13.3	
Air pollution (PM 2.5)		0.6	10.5		−8.8	17.7		−27.9	17.4	
**Covariates**										
% non-Latinx Black residents		59.3	31.7		−21.7	48.2		5.1	34.3	
% non-Latinx Asian residents		−13.3	11.4		−70.5	39.5		−24.0	19.9	
**Unexplained**		406.9	104.0	36.7	491.7 *	198.2	37.3	247.0 *	122.4	41.2

Notes. * *p* < 0.05, ** *p* < 0.01, and *** *p* < 0.001. B (coefficient) and SE (standard error) are reported. The robustness tests concerning only municipalities that reported data across all three waves (*n* = 368) reported robust results for the majority of the factors. Changes are as follows: in Wave 1, % less than high school education became a marginally significant factor; in Wave 2, household crowding and % Trump voters became marginally significant factors. a. Differences in exposure and vulnerability factors between RHL and RLL are reported with significance level of t-tests adapted from Table A2. No statistically significant spatial autocorrelation was identified for Wave 1 infection rates after controlling for all predictors. In models with a spatially lagged dependent variable as a predictor, % Trump voters became marginally significant in Wave 2, and % Carpool or public transportation became statistically significant at *p* < 0.05 level in Wave 3. ^a^ differences in exposure and vulnerability factors between RHL and RLL are reported with significance level of *t*-tests adapted from Table A2.

**Table 3 ijerph-19-13963-t003:** Decomposition analysis contrasting RHB to RLB municipalities.

	Diff (RHB-RLB) ^a^	Wave 1 (*n* = 558)			Wave 2 (*n* = 406)		
		B	SE	% Diff	B	SE	% Diff
Infection rates (RHB)		1558.4	79.8		4700.7	119.4	
Infection rates (RLB)		1254.9	69.8		4353.3	84.7	
Difference		303.6 **	106.0	100.0	347.4 *	146.4	100.0
**Explained**		254.4 *	114.7	83.8	246.2	178.0	70.9
**Exposure factors**							
Household crowding	1.50 ***	29.9	41.2		159.3 *	67.9	45.9
Household size		47.1	25.1		−31.4	24.2	
% Healthcare workers		15.3	16.7		67.4	35.6	
% Public safety workers		−18.9	17.0		28.3	27.9	
% Food industry workers		−20.9	16.2		11.5	24.8	
% Transportation workers	1.74 ***	3.3	38.3		203.7 **	65.2	58.6
% Utilities workers		−4.3	7.1		−4.8	10.2	
% Carpool or public transportation		4.0	21.9		64.0	42.5	
Population density		2.9	12.4		−28.3	24.0	
% Urbanized areas		−13.2	11.7		35.7	22.7	
Distance to NYC	9.22 ***	−152.2 **	50.0	−50.1	124.9 *	58.9	35.9
Stability		−8.9	27.1		−115.6	66.6	
Long-term care beds		27.0	16.1		25.1	18.5	
% Trump voters	−14.83 ***	23.1	66.2		−497.9 ***	133.9	−143.3
**Vulnerability factors**							
All-cause mortality rate		−95.1	64.1		−26.1	23.5	
% 65+		22.8	39.4		106.9	67.5	
Acute care facilities		−24.3	18.6		16.0	23.5	
Gini		1.8	5.9		12.2	22.7	
% English less than “very well”		−25.0	46.7		−14.8	36.3	
Per capita income		−92.1	86.4		−179.0	165.3	
% Less than high school		405.9	212.1		−86.8	107.2	
% Unemployed		−72.3	38.9		−78.9	61.1	
Air Pollution (PM 2.5)		4.9	30.1		8.6	54.9	
**Covariates**							
% Latinx residents	8.42 ***	192.5 **	73.8	63.4	437.2 ***	125.6	125.9
% non-Latinx Asian residents		1.4	4.4		9.2	14.3	
**Unexplained**		49.1	98.5	16.2	101.1	173.5	29.1

Notes. * *p* < 0.05, ** *p* < 0.01, and *** *p* < 0.001; B (coefficient) and SE (standard error) are reported. The robustness tests with only data from municipalities that reported data for all three waves (*n* = 368) reported robust results for all exposure and vulnerability factors. Changes are found for % Latinx residents. In Wave 1, % Latinx residents became a non-significant factor. No statistically significant spatial autocorrelation was identified for Wave 1 infection rates after controlling for all predictors. In models with spatially lagged dependent variable as a predictor, distance to NYC became marginally significant in Wave 2. ^a^ differences in exposure and vulnerability factors between RHB and RLB are reported with significance level of t-tests adapted from Table A4.

## Data Availability

The data presented in this study are available on request from the corresponding author.

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
