# Peer review of "Municipal Ethnic Composition and Disparities in COVID-19 Infections in New Jersey: A Blinder–Oaxaca Decomposition Analysis"

_ijerph, 2022, doi:10.3390/ijerph192113963_

Round 1
Reviewer 1 Report
Please see the attached document for detailed comments.

Reviewer 2 Report
The study is very interesting with a strong methodology. It has minor flaws regarding the numbering sequence of the figures (e.g., Figure 4 title says Figure 2, lines 307-308).
Introduction: It does have originality, theoretical and conceptual background, and a good literature review.
Material and methods: Strong study design and valuable statistical analysis.
Results. Clear and understandable. Please review the figure title numbering sequence.
Discussion and conclusions. OK
I really enjoy reading your work. Congratulations.
Author Response
Thank you very much for your support and enthusiasm of this manuscript. We have corrected and double checked the numbering sequencing of all of our tables and figures.
Round 2
Reviewer 1 Report
Thank you, sincerely, the authors for this thoughtful response to review and revision of the manuscript. I feel that my concerns have been sufficiently addressed and appreciate the input of a qualified statistician on the choice of analysis. I recommend some minor editing for agreement and diction, but otherwise, look forward to seeing this piece published!